# Social dominance mediates behavioral adaptation to chronic stress in a sex-specific manner

**Stoyo Karamihalev[1,2†], Elena Brivio[1,2†], Cornelia Flachskamm[1], Rainer Stoffel[1], Mathias V Schmidt[1], Alon Chen[1,3]***

[1]Department of Stress Neurobiology and Neurogenetics, Max Planck Institute of Psychiatry, Munich, Germany; [2]International Max Planck Research School for Translational Psychiatry (IMPRS-TP), Munich, Germany; [3]Department of Neurobiology, Weizmann Institute of Science, Rehovot, Israel

**Abstract** Sex differences and social context independently contribute to the development of stress-related disorders. However, less is known about how their interplay might influence behavior and physiology. Here we focused on social hierarchy status, a major component of the social environment in mice, and whether it influences behavioral adaptation to chronic stress in a sex-specific manner. We used a high-throughput automated behavioral monitoring system to assess social dominance in same-sex, group-living mice. We found that position in the social hierarchy at baseline was a significant predictor of multiple behavioral outcomes following exposure to chronic stress. Crucially, this association carried opposite consequences for the two sexes. This work demonstrates the importance of recognizing the interplay between sex and social factors and enhances our understating of how individual differences shape the stress response.

***For correspondence:**
Alon.Chen@weizmann.ac.il

[†]These authors contributed equally to this work

**Competing interests:** The authors declare that no competing interests exist.

## Introduction

Stress-related psychopathologies, such as mood and anxiety disorders, show a pronounced gender bias in their prevalence, severity, age-of-onset, and most common comorbidities (*Altemus et al., 2014*; *Bangasser and Valentino, 2014*; *Kessler et al., 2005*; *Young and Pfaff, 2014*). For example, the latest studies estimate the prevalence of major depressive disorder among women as 1.5 times higher than in men (*World Health Organization, 2017*). In addition, in women major depression is characterized by increased symptom severity (*Martin et al., 2013*) and is more commonly comorbid with anxiety disorders, eating disorders, and sleep disturbances, while men with major depression are more prone to develop aggression, alcohol or substance abuse, and suicidal ideation (*Marcus et al., 2005*; *Martin et al., 2013*).

Despite these observations and the documented examples of sexual dimorphism in human stress response (*Bangasser and Valentino, 2012*), the biological mechanisms that give rise to sex differences in stress response are not well understood (*Beery and Zucker, 2011*; *Joel and McCarthy, 2017*). The symptomatology of stress-related pathologies and the biological response to stress span several domains of functioning including energy metabolism, mood, and sociability. Recent studies in rodent models of stress-related psychopathologies have already identified several differences across molecular, behavioral, and metabolic levels (*Bangasser and Wicks, 2017*; *Brivio et al., 2020*; *Hodes, 2018*; *Hodes and Epperson, 2019*; *Young and Pfaff, 2014*). Very few studies, however, have looked into the interaction between pre-existing differences in social behavior between the sexes and stress. Considering that abnormalities in social functioning are an essential part of the symptomatology of stress-related disorders, differences in social behavior and social cognition prior to disorder onset are likely to contribute to disorder susceptibility. Here, we explored how social

**eLife digest** Most people experience chronic stress at some point in their life, which may increase their chances of developing depression or anxiety. There is evidence that chronic stress may more negatively impact the well-being of women, placing them as higher risk of developing these mental health conditions. The biological factors that underlie these differences are not well understood, which leaves clinicians and scientists struggling to develop and provide effective treatments.

The social environment has a powerful influence on how people experience and cope with stress. For example, a person's social and socioeconomic status can change their perception of and reaction to everyday stress. Researchers have found differences in how men and women relate to their social standing. One way for scientists to learn more about the biological processes involved is to study the effect of social standing and chronic stress in male and female mice.

Now, Karamihalev, Brivio et al. show that social status influences the behavior of stressed mice in a sex-specific way. In the experiments, an automated observation system documented the behavior of mice living in all female or male groups. Karamihalev, Brivio et al. determined where each animal fit into the social structure of their group. Then, they exposed some groups of mice to mild chronic stress and compared their behaviors to groups of mice housed in normal conditions. They found that both the sex and social status of each played a role in how they responded to stress. For example, subordinate males displayed more anxious behavior under stressful circumstances, while dominant females acted bolder and less anxious.

More studies in mice are needed to understand the biological basis of these social- and sex-based differences in stress response. Learning more may help scientists understand why some individuals are more susceptible to the effects of stress and lead to the development of personalized prevention or treatment strategies for anxiety and depression.

context shapes the response to chronic stress. We focused specifically on social dominance, an essential characteristic of rodent social groups.

Wild and laboratory rodents form complex and dynamic social structures which typically involve the formation of dominance hierarchies (*Kondrakiewicz et al., 2019*). These have been observed in the lab in group sizes ranging from three to over a dozen individuals (*Horii et al., 2017*; *Varholick et al., 2019*; *Wang et al., 2014*). Hierarchies are thought to improve social stability and reduce severe conflicts and aggression (*Curley, 2016*). As a consequence, an individual's position in the dominance hierarchy has important consequences, including preferential access to food, shelter, and mates (*Drews, 1993*). Social rank within male hierarchies is also known to influence health, hormonal profile, brain function, metabolism, and mortality (*Pallé et al., 2019*; *Razzoli et al., 2018*). For instance, subordinate individuals display increased anxiety-like behavior, a suppressed immune response, higher basal corticosterone levels, and reduced life span (*Bartolomucci, 2007*). These types of relationships have classically been studied in male animals, as female mice have usually appeared more communal and displayed limited aggression (*König and Lindholm, 2012*). Recent work, however, has demonstrated that female laboratory mice also form hierarchies that appear quite similar to those seen in males, accompanied by some of the same dominance-related physiological markers, such as differences in corticosterone levels (*Schuhr, 1987*; *van den Berg et al., 2015*; *Varholick et al., 2019*; *Varholick et al., 2018*; *Williamson et al., 2019*). Thus, we examined social dominance status as a putative mediator of sex differences in the response to adverse events.

To do so, we took advantage of a high-throughput automated behavioral monitoring system (the Social Box, SB) to assess and better understand the hierarchies of groups of male or female mice (*Forkosh et al., 2019*; *Shemesh et al., 2013*). We then exposed mice to a well-established chronic stress procedure, the chronic mild stress (CMS) paradigm, and evaluated its effects using a series of standard behavioral and physiological readouts. Finally, we used social dominance status at baseline to predict behavioral outcomes following CMS. We hypothesized that an individual's standing in the social hierarchy would be a predictor of behavior upon stress exposure, and that this relationship would differ between the sexes.

## Results

### Male and female dominance hierarchies

We first explored the hierarchical structure of grouped CD-1 mice over four days of baseline monitoring as well as the stability of hierarchies following an acute stressor (15 min of restraint stress). Social dominance was assessed by calculating the David's Score (DS), an established method for inferring social hierarchies (*David, 1987*; *Gammell et al., 2003*). We based the DS on the numbers and directionality of chases between each pair of individuals in a group. A cumulative DS for the four baseline days of the SB assessment was used as a final measure of social dominance. In line with previous studies (*Schuhr, 1987*; *van den Berg et al., 2015*; *Varholick et al., 2019*; *Varholick et al., 2018*; *Williamson et al., 2019*), we were able to detect some stability in the hierarchies of both sexes (*Figure 1a*).

We further calculated several properties of male and female hierarchies to explore potential differences in their characteristics. Namely, we calculated: (1) steepness – a measure of social distance between each individual in the hierarchy, (2) despotism – a measure of the extent to which the top-ranking individuals dominate over the rest of the group, (3) directional consistency – the extent to which the directionality of the interactions follow the expected direction from higher to lower rank, and (4) Landau's modified $h'$ – a measure of hierarchy linearity (*de Vries, 1995*; *Landau, 1951*; *Figure 1—figure supplement 1*). We found that male hierarchies were steeper, more linear, more despotic, and had higher directional consistency than those of females. Interestingly, mice housed in larger groups show analogous relationships between sexes (*Williamson et al., 2019*).

To investigate if social rank carries any sex-specific implications for overall behavior, we analyzed the correlation structure between an individual's DS and 59 behavioral readouts recorded post-habituation (days 2–4) in the SB within each sex (briefly described in Methods. For a detailed list of behaviors and how they are computed, see *Forkosh et al., 2019*). Thirty of the fifty-nine behavioral readouts tested (50.84%) showed significant correlations with cumulative baseline DS in at least one of the sexes (*Figure 1—figure supplement 2*, $q < 0.05$, Spearman's rank correlation, Benjamini-Hochberg adjustment within each sex). While the overall association pattern was quite similar between male and female mice, there were several correlations seen in males that were absent in females (*Figure 1b* and *Figure 1—figure supplement 2*). These included measures of overall locomotion, such as *Distance Outside* (Male $r_s = 0.478$, $n = 40$ mice, $p = 0.00355$. Female $r_s = -0.253$, $n = 48$ mice, $p = 0.137$), and *Fraction of Time Outside* – the mean proportion of time a mouse spent outside the nest (Male $r_s = 0.523$, $n = 40$ mice, $p = 0.00125$. Female $r_s = -0.144$, $n = 48$ mice, $p = 0.402$), as well as two related measures of roaming entropy, which assess the predictability of how an individual explores their environment – *Entropy* and *Grid Entropy [6 × 6]* (for brevity we only report the latter, Male $r_s = 0.531$, $n = 40$ mice, $p = 0.00123$. Female $r_s = 0.00677$, $n = 48$ mice, $p = 0.694$). These correlations indicate that overall locomotion and exploration of the home environment may be more strongly connected to social rank in male groups, while being seemingly independent of social status in females. Interestingly, no correlations were present in females but absent in males. Altogether these findings suggest that male and female social dominance hierarchies, despite having a similar structure, have different relationships to overall behavior.

Next, we estimated DS stability over time by examining the frequency of rank change events and comparing those to the chance-level expectation. Briefly, normalized daily DS values were ranked for each group to create a four-rank hierarchy: $\alpha$ (most dominant), $\beta$, $\gamma$, and $\delta$ (most subordinate) and each mouse was assigned a single rank based on its four-day cumulative DS. For each pair of consecutive days, we observed how many individuals maintained the same rank they had been assigned on the previous day. We then calculated the rank maintenance odds for animals in each final rank category relative to the expected chance-level (*Figure 1c*). The true probability of rank maintenance in our data was higher than chance in $\alpha$-females and in all male ranks (one-tailed binomial tests against the rank maintenance probability of 25%, $\alpha$-females: 21/36 successes, $p = 2.1 \times 10^{-5}$, $\alpha$-males: 22/30 successes, $p = 3.7 \times 10^{-8}$, $\beta$- and $\gamma$-males: 13/30 successes each, $p = 0.0216$, $\delta$-males: 19/30 successes, $p = 1.02 \times 10^{-5}$). These results indicate that the highest rank in a hierarchy is often occupied by the same individual over time in both sexes, while the lower ranks appeared to be stable in males only (*Figure 1c*).

In addition to stability over time during baseline recordings, individual DS also remained stable following acute restraint stress (Pearson's correlation between cumulative baseline DS and DS

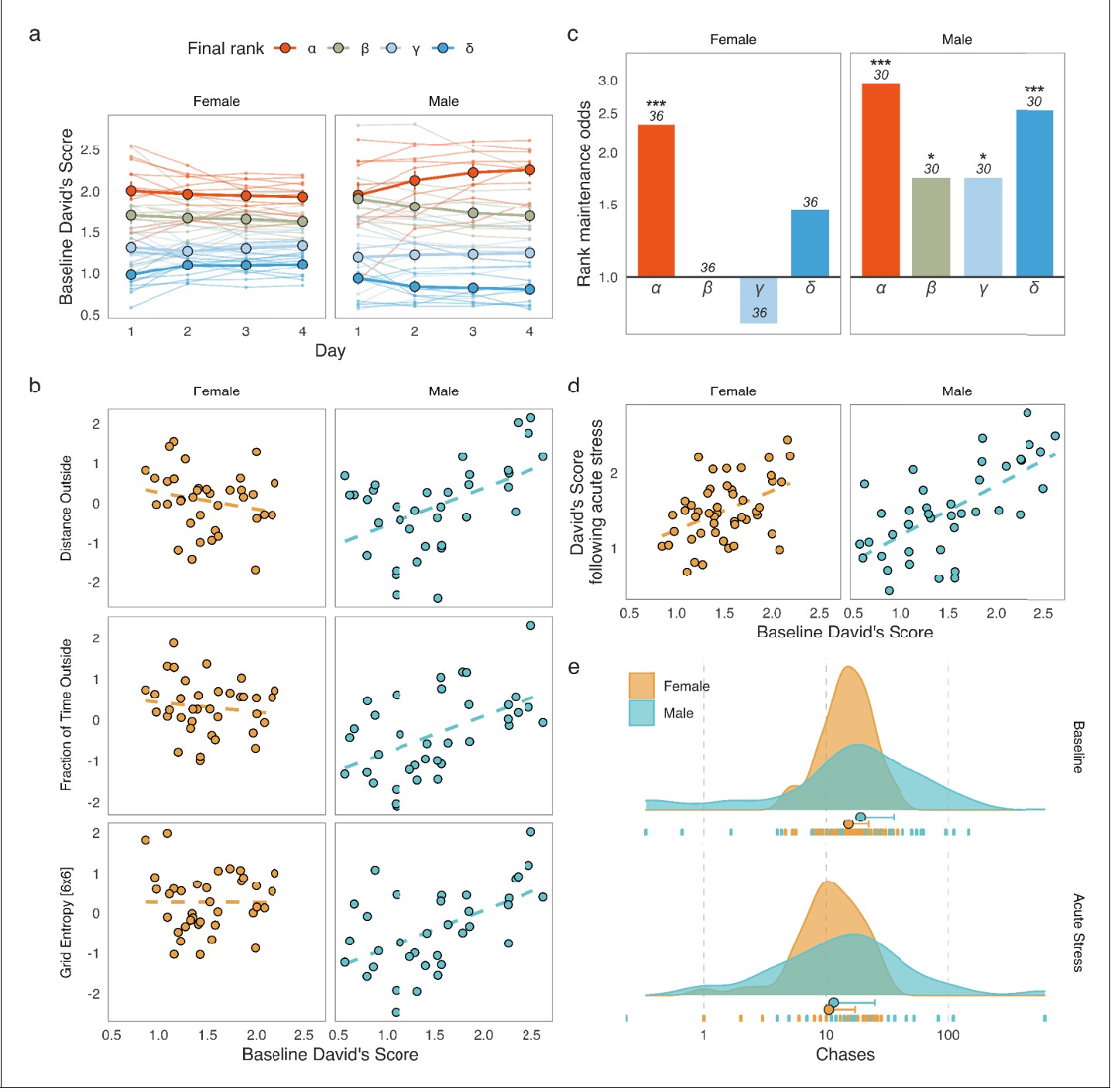

**Figure 1.** Social dominance hierarchies in males and females. (**a**) David's Scores (DS) based on chases during the four baseline days of Social Box (SB) assessment show relatively stable social hierarchies in both male and female groups (each line represents an individual, colors represent the cumulative social rank on day 4, points are mean values for each rank ± standard error of the mean). (**b**) Male-specific associations between social dominance scores and behaviors related to locomotion and exploration. Dominant males had increased overall locomotion, spend more time outside the nest, and moved through the SB environment in a more unpredictable manner. These associations were not found in females. (**c**) Rank maintenance odds over the four-day baseline period. Depicted are odds of maintaining the same rank between consecutive days relative to chance-level (25%). Data is summarized according to the cumulative social rank on day 4; numbers indicate the number of individuals per rank. (**d**) Baseline DS predicts DS following acute restraint stress in both sexes, indicating that social dominance hierarchies may be relatively robust against acute stress. (**e**) Numbers of chases in male and female groups at baseline as well as following acute restraint stress. Both sexes display significantly fewer chases following an acute physiological stressor. The x-axis shows the absolute number of chases between pairs of mice. Dot: median, whisker: 1.5 x IQR.

The online version of this article includes the following figure supplement(s) for figure 1:

*Figure 1 continued on next page*

**Figure supplement 1.** Social hierarchy properties in male and female groups.

**Figure supplement 2.** Associations between David's Scores (DS) and other behaviors measured in the Social Box (SB).

following acute restraint on day 5, *Figure 1d* and *Figure 1—figure supplement 1*). Both males ($r = 0.6639$, $n = 40$ mice, p = $3 \times 10^{-6}$) and females ($r = 0.446$, $n = 48$ mice, p = 0.000149) showed significant DS correlations from baseline to acute restraint. Finally, we investigated the possibility of differential effects of acute restraint on the behavior used to produce the DS – numbers of chase events (*Figure 1e*). Repeated-measures ANOVA on log-transformed chase numbers showed that the number of chases decreased significantly following acute restraint stress ($F(1, 86) = 29.04$, $n = 88$ mice, p = $6.11 \times 10^{-7}$), however the extent of this decrease did not differ between the sexes (Sex x Stage interaction, $F(1, 86) = 1.053$, $n = 88$ mice, p = 0.301).

The apparent robustness of social hierarchies over time and in response to acute stress suggested that predictions from the baseline assessment may carry information that would still be relevant to behavioral outcomes following a long-term intervention. More specifically, we hypothesized that occupancy of the highest-ranking positions in the social hierarchy in both sexes and additionally the lowest in males might be sufficiently stable to allow for long-term predictions.

## Effects of CMS on behavior and physiology

To investigate the effects of pre-existing social dominance status on the behavioral response to chronic stress, we employed a CMS protocol adapted for group-housed animals.

In short, groups were exposed to a weekly schedule of two daily randomly combined mild stressors (e.g. wet bedding, tilted cage, overcrowding) for a total of three weeks. Six groups of each sex ($n = 24$ per sex) were randomly assigned to receive CMS, while the rest of the groups (six groups of females and four groups of males) were assigned to the control condition. The 21-day CMS procedure was followed by a behavioral test battery for both control and CMS mice, which included tests previously shown to capture the effects of chronic stress (*Figure 2a*). This included, among others, classical tests of locomotion (open field test, OFT), anhedonia (sucrose preference test, SPT), anxiety-like behavior (elevated plus maze, EPM), and stress coping (tail suspension test, TST). Additionally, we assessed several physiological indicators of stress level (*Figure 2b–e*). All the physiological and behavioral outcome variables following CMS were collected into a single dataset. Since the full experiment was run in two batches, all outcome variables were adjusted for batch effect (*see* Methods). To improve readability, we report the batch-adjusted values relative to the mean of female control mice.

As expected, we found that both bodyweight change and cumulative coat quality were significantly reduced following CMS in both males and females (*Figure 2b–c*, Bodyweight: $F(1, 82) = 7.394$, p = 0.00798, Coat quality: KW test, $\chi^2(1) = 18.586$, p = $1.6 \times 10^{-5}$), although post-hoc pairwise comparisons indicated a bodyweight difference in females only (females: $t(43.784) = 3.9447$, p = 0.000285, males: $t(36.937) = 1.1064$, p = 0.27). Bodyweight-adjusted adrenal weights were increased after CMS in males only (*Figure 2d and* -way ANOVA, sex by condition interaction, $F(1, 80) = 4.42$, p = 0.039, followed by pairwise within-sex 2-sided t-tests: males: $t(27.03) = -3.143$, p = 0.004; Females: $t(41.18) = 0.0726$, p = 0.94). For all further analyses, these physiological outcomes were combined with the behavioral ones in a single dataset.

## Sex-specific effects of dominance on CMS outcomes

To explore how exposure to chronic stress shapes behavior in groups of mice, we investigated the major drivers of variance in the dataset containing all behavioral and physiological readouts following CMS using principal components analysis (PCA, *Figure 3a–d*). The first principal component (PC1), explained approximately 21.6% of the variance in the outcome data (*Figure 3a*). To our surprise, neither sex nor condition (CMS vs controls) appeared to capture variance contained in PC1 (*Figure 3b*, condition effect: $F(1, 82) = 0.608$, p = 0.44). Instead, sex and condition were associated with PC2 and PC3 respectively (*Figure 3—figure supplement 1*). Since none of the expected variables (sex, condition, or their interaction) contributed to the main source of variance in the dataset, we investigated whether social dominance was a contributing factor. We tested the association

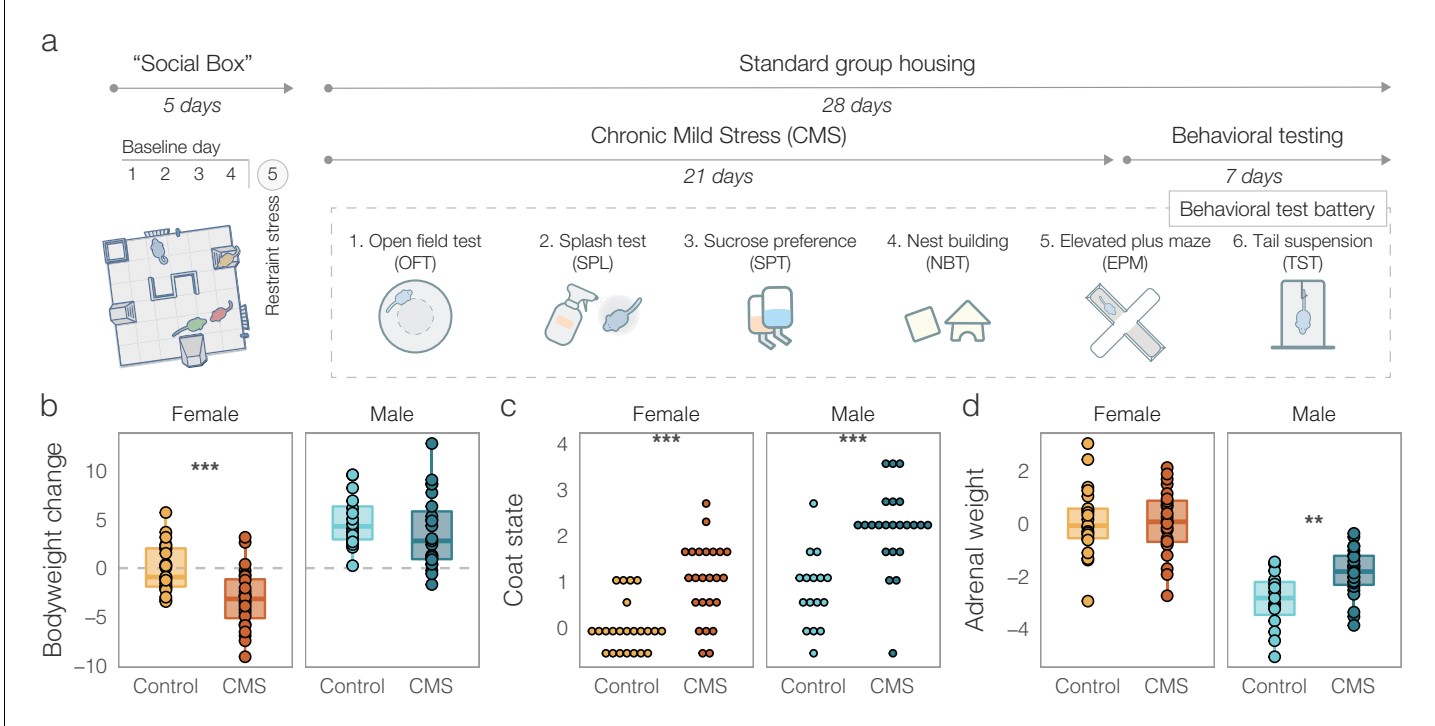

**Figure 2.** Outcomes of chronic mild stress (CMS) in males and females. (a) Experimental timeline. All groups underwent five days of Social Box (SB) monitoring. This consisted of four days of baseline monitoring followed by a 15-min acute restraint stress for all individuals prior to being re-introduced into the SB for a final 12-h dark phase monitoring period (day 5). After the SB, groups received three weeks of either control treatment (bodyweight and fur quality assessments two times a week) or CMS (see Materials and methods for details). The following week, all groups underwent a behavioral test battery in the order depicted. (b) Batch-adjusted bodyweight change following three weeks of CMS. Both male and female CMS mice showed significantly reduced weight compared to controls. (c) Batch-adjusted coat state scores (higher means poorer fur quality) following CMS. Male and female CMS groups showed significant deterioration of their coat. (d) Batch- and initial bodyweight-adjusted adrenal weights. CMS increased adrenal size in males, but not in females. Boxplots: line – median, box limits – 1st and 3rd quartile, whiskers – 1.5 x IQR. Data is presented relative to female controls. Number of mice per condition: Female Control = 23, Female CMS = 24, Male Control = 16, Male CMS = 23. (*p < 0.05, **p < 0.01, ***p < 0.001).

between PC1 scores and DS (*Figure 3c*). Remarkably, baseline DS significantly predicted scores on PC1 in CMS individuals only and this association was in opposite directions between the two sexes (sex by DS interaction: $F(1, 43) = 6.016$, $p = 0.0183$). Thus, the principal source of variation in the outcome dataset contained an interaction between baseline dominance scores and sex in the CMS mice.

To better assess the set of behaviors responsible for this association, we correlated PC1 scores with all the input features from the behavioral and physiological readouts (*Figure 3d*). We found that seventeen readouts were significantly correlated with PC1 scores in this dataset (Spearman's rank correlation, Bonferroni-adjusted p < 0.05). Among the strongest correlates of PC1 were measures derived from the OFT and EPM, and specifically features related to locomotion and anxiety-like behavior, such as distance traveled and visits to the anxiogenic regions of test chambers. Interestingly, these behaviors do not typically differentiate CMS and control individuals. Instead, CMS exposure appeared to create relationships between dominance and the outcome variables that were not present in controls (the top examples from the OFT and EPM are depicted in *Figure 3e–f*, correlations between individual readouts and DS within each sex and condition are available in *Figure 3— figure supplement 2*). To conclude, we were able to narrow down a portion of the variance in a broad range of behavioral and physiological outcomes following CMS to an interaction between dominance and sex with more subordinate CMS males showing apparent increases in measures of overall activity (distance/speed in the OFT and EPM) and more subordinate males and more dominant females showing an apparent reduction in anxiety-like behavior. Thus, we were able to identify

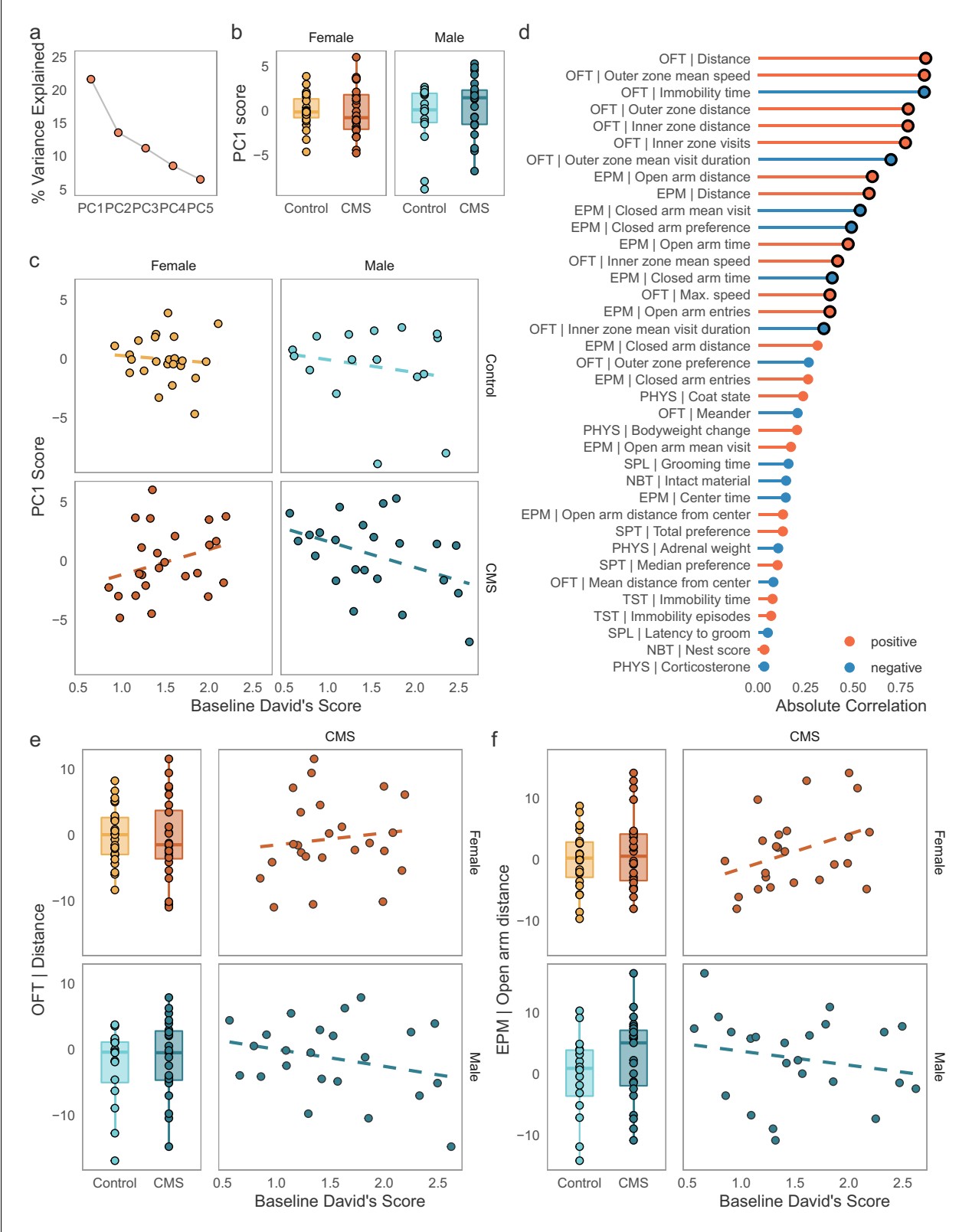

**Figure 3.** Opposing effects of baseline social dominance scores on behavioral outcomes following CMS. (**a**) Percentage of variance explained by the first five components of a principal components analysis conducted on the batch-adjusted behavioral and physiological outcome data. PC1 explains ca. 21% of the variance in this dataset. (**b**) PC1 is not significantly different between sexes or conditions, indicating that this component did not capture variance associated with either variable. (**c**) Association between baseline David's Scores and PC1 in control and CMS individuals. Baseline dominance

*Figure 3 continued on next page*

*Figure 3 continued*

predicted scores on PC1, the major source of variance in the outcome data, in a sex-specific manner in the CMS group, but not in the control group. (**d**) Spearman's rank correlations between PC1 and the physiological and behavioral outcome variables. The strongest associations for PC1 are variables derived from the open field test (OFT) and elevated plus maze (EPM). Black circles around points identify associations significant at $p < 0.05$ after adjustment for multiple testing (Bonferroni correction). (**e-f**) Examples of interactions between baseline dominance and sex on CMS behavioral outcomes. Males and females show significant opposite correlations between dominance and example of locomotion and anxiety-like behavior. (Boxplots: line – median, box limits – 1st and 3rd quartile, whiskers – 1.5 x IQR. Scales for behavioral outcomes are relative to female controls).

The online version of this article includes the following figure supplement(s) for figure 3:

**Figure supplement 1.** PC2 and PC3 capture variance associated with sex and condition.

**Figure supplement 2.** Correlations between behaviors and baseline dominance scores.

a novel role for social rank belonging in modulating behavior following chronic stress in a sexually dimorphic way.

## Discussion

Social behavior in general and social dominance in particular are important contributors to individual differences (*Forkosh et al., 2019*). As such, they may also shape how individuals respond to environmental challenges. In humans, different types of social hierarchies coexist in complex structures and they influence an individual's behavior and health (*Sapolsky, 2005*). Mouse social dominance hierarchies are considerably simpler, however parallels between the two species can be drawn in particular with regard to human socioeconomic status (SES). Both objective and perceived SES impact human health, mortality, morbidity, and susceptibility to psychiatric disorders such as depression or anxiety (*Farah, 2017*; *Freeman et al., 2016*; *Hoebel et al., 2017*; *McEwen and Gianaros, 2010*; *Shaked et al., 2016*; *Wetherall et al., 2019*). These outcomes resemble findings related to social dominance in mice (*Bartolomucci, 2007*; *Pallé et al., 2019*; *Razzoli et al., 2018*). Intriguingly, some human studies have identified sex differences in the relationship between SES and stress-related psychopathologies (*Kosidou et al., 2011*; *Mwinyi et al., 2017*; *Peplinski et al., 2018*). All this suggests that social status and the stress and health gradient that characterize human social structures (*McEwen and Gianaros, 2010*) likely have close analogues in other mammals with well-defined social structures.

Here, we have demonstrated that both male and female socially housed mice establish social dominance hierarchies, which are relatively stable over time and resistant to acute perturbations. In agreement with previous work, female hierarchies were less despotic and had lower directional consistency (*Williamson et al., 2019*), suggesting that females may be maintaining a less rigid structure compared to males. This is supported by our finding that only the top rank in females showed significant stability over time, whereas in males both subordinate and dominant ranks appeared stable. Further research is needed to ascertain if this observation is limited to our paradigm and specific set of measurements (same-sex groups of four individuals), or if it represents a true sex difference in social dominance hierarchies.

Additionally, our data suggest that an individual's position in the hierarchy carries different implications for overall behavior in each sex. While we are unable to assess whether social rank belonging should be considered a cause or consequence of these behavioral differences, we have observed an interesting sex difference in this relationship. For groups living in the enriched environment of the SB apparatus, dominance in males but not in females was associated with overall locomotion, proportion of time spent outside the nest, and exploration entropy. These associations likely reflect territorial or patrolling behavior in males, which may be less relevant to female social hierarchies.

As hypothesized, occupancy of different positions in the social hierarchy conferred varying levels of responsiveness to the challenges posed by chronic stress. Previous investigations have rarely found associations between social dominance and response to chronic stress (*Larrieu and Sandi, 2018*). An important exception is a recent study by *Larrieu et al., 2017* in groups of male mice exposed to chronic social defeat stress. The authors found increased susceptibility to chronic social defeat for dominant males, but in contrast to our results, no real behavioral alterations for subordinates. It is important to note, however, that chronic social defeat and CMS are profoundly different paradigms. Social defeat is strongly tied to social dominance and might be perceived as loss of

status more than other stressors (*Larrieu and Sandi, 2018*). Our use of CMS allowed us to investigate both sexes under comparable levels of stress. Nevertheless, both the study from Larrieu and colleagues, and ours highlight that social status can influence an individual's response to long-term adverse life events.

Importantly, we demonstrated that the effects of preexisting dominance on stress outcomes were sexually divergent, such that the association between dominance and anxiety-like and locomotor behavior following CMS was in opposite directions between males and females. Specifically, subordinate males appeared to display hyperlocomotion, while dominant females displayed increased boldness (reduced anxiety-like behavior) compared to non-CMS controls. Overall, our data indicate that an individual's position within a social structure can influence their behavioral response to chronic stress in a sex-specific fashion.

These findings suggest an intriguing possibility. Given that male social hierarchies are likely antagonistic, we speculate that social living carries an especially high cost for subordinate males, who are the recipients of most antagonistic interactions. Conversely, female hierarchies may contribute to more affiliative social interactions, and thus social context may carry a net benefit for females, with the highest benefit gained by the dominant females. We speculate that this positioning as the most advantaged and disadvantaged individuals may confer higher behavioral flexibility and results in the strongest behavioral change upon exposure to environmental challenges.

Crucially, since we decided to maintain social context throughout our experimental design, the current work did not allow for the assessment of the effect of group- versus single-housing on CMS outcomes. Given this constraint, we were not able to confidently assess the difference in how CMS was experienced by each sex in groups as opposed to if they had been single-housed. However, since we were interested in the prediction from baseline dominance, we did not wish to remove the salience and thereby the effect of social context. Likewise, in naturalistic conditions, mice are found in mixed-sex groups (*Kondrakiewicz et al., 2019*). Working with same-sex groups provided us with a more controlled environment, preventing confounding by mating behavior and pregnancy. However, this was at the expense of the ethological validity of our findings. Further research is needed to understand if and how mixed-sex social structures may differ in their impact on stress outcomes.

Moreover, while CMS produced some of the expected physiological changes (i.e., reduction of bodyweight gain, reduced coat quality, adrenal weight increase), we did not observe several of the behavioral phenotypes often found using similar protocols (e.g., hyperlocomotion, anhedonia, passive coping, *Franceschelli et al., 2014*). While we have sufficient evidence that CMS individuals experienced significant amounts of stress, we are not able to determine if the absence of some of these behavioral signatures of CMS is a result of the maintenance of social context throughout the protocol or if it is due to other unknown factors. We are, however, not the first to observe no change in adrenal size or sucrose preference in female CD1 mice (*Dadomo et al., 2018*). Additionally, we did not observe any changes in basal corticosterone levels. This is probably be due to the fact that our blood sampling was performed one week after the end of the CMS paradigm, allowing enough time for corticosterone levels to return to normal.

Finally, we employed the David's Score as a continuous linear indicator of social dominance for the additional statistical power that this approach provides. Dominance hierarchies, however, are more commonly thought of as ordinal, and we lacked sufficient sample sizes per rank and condition to be able to reliably quantify the contribution of each rank to the behavioral outcomes of chronic stress. Further research is needed to replicate and extend these findings to specific social ranks.

While were not able to directly compare between single- and group-housed animals, our data suggest that the existence of a social hierarchy in groups of mice might contribute to increased variability in behavioral outcomes after chronic treatment generating rank-specific responses. Moreover, this effect could be especially relevant when studying sex differences. Often, housing conditions (single vs. group) are not taken into consideration as a variable of interest. Based on the findings reported here, we speculate that housing conditions might have contributed to discordant behavioral findings in studies of stress and sex (*Franceschelli et al., 2014*). Our results argue for considering group-derived individual differences and, in particular, dominance status, in the design of experiments, especially when investigating the contribution of sex differences to stress response.

Taken together, this work suggests that social dominance might influence the perception of and reaction to chronic stress differently for male and female mice. While there has been some work looking into the effects of dominance on stress susceptibility in males (*Larrieu et al., 2017*), very

little is known about female social dominance and its contribution to stress coping. Our findings emphasize the need for exploring the stress response in the presence of conspecifics in a more naturalistic manner and the importance of recognizing that the same social factors may carry divergent consequences for the behavior of males and females.

# Materials and methods

## Key resources table

| Reagent type (species) or resource | Designation | Source or reference | Identifiers | Additional information |
|---|---|---|---|---|
| Commercial assay or kit | Corticosterone Double Antibody RIA Kit | MP Biomedicals | SKU 0712010-CF | |
| Software, algorithm | *Tidyverse*, ecosystem of packages | *R Core Development Team, 2013* doi:10.21105/joss.01686 | | R version 4.0.2 |
| Software, algorithm | 'RNOmni R package | *McCaw, 2019* | v 0.7.1 | R version 4.0.2 |
| Software, algorithm | 'cowplot' R package | *Wilke, 2019* | v 1.0.0 | R version 4.0.2 |
| Software, algorithm | 'compete' R package | *Curley et al., 2015* | v 0.1 | R version 4.0.2 |
| Software, algorithm | 'steepness' R package | *Leiva and de Vries, 2014* | v 0.2–2 | R version 4.0.2 |

## Animal housing and care

Male and female ICR CD-1 mice at 7–9 months old were employed for all experiments (Charles River, Sulzfeld, Germany). Mice were housed in groups of four in the animal facilities of the Max Planck Institute of Psychiatry in Munich, Germany, from weaning and were maintained under standard conditions (12L:12D light cycle, lights on at 07:00 AM, temperature 23 ± 2°C) with food and water available ad libitum. All experiments were approved by and conducted in accordance with the regulations of the local Animal Care and Use Committee (Government of Upper Bavaria, Munich, Germany), under licenses Az.: 55.2-1-54-2532-148-2012, Az.:55.2-1-54-2532-32-2016 and ROB-55.2–2532.Vet_02-18-50. The fur of all mice was marked using four different colors under mild isoflurane anesthesia and mice were left to recover for several days before the start of the experiment. On day 1, animals were transferred to the SB (see 'The 'Social Box' paradigm' section), for a total of 5 days (five light periods and five dark periods). On day 6, animals were removed from the SB and placed in their original cage under standard housing conditions for the rest of the experimental procedure (see 'Chronic mild stress protocol' and 'Behavioral battery' sections).

## Behavior in a semi-naturalistic environment

### The 'Social Box' paradigm

The 'Social Box' is a behavioral arena wherein groups of mice live under continuous observation over a period of several days (*Forkosh et al., 2019*; *Shemesh et al., 2013*). Mouse identities are maintained using fur markings in four different colors (*Shemesh et al., 2020*; *Forkosh et al., 2019*). The entire SB observation period was recorded using cameras mounted above each arena. Videos of the dark periods of the light cycle (7:00 PM to 7:00 AM) were then compressed and analyzed using a custom automated tracking system which determines mouse locations over time based on color segmentation (*Shemesh et al., 2013*; *Shemesh et al., 2020*). From the location data we inferred agonistic interactions as well a variety of other behavioral readouts as described in *Forkosh et al., 2019*. Briefly, we used the absolute locations and smoothed movements of individuals as well as location and movement with respect to regions of interest in the SB to compute readouts related to overall locomotion, feeding/drinking, etc. (e.g., distance and speed outside the nest, distance from walls,

distance from the nest, time spent in the feeders, on ramps, in the S-wall). Exploratory behavior is assessed by estimating the unpredictability of movement (entropy) outside the nest using spatial bins of either the regions of interest or a 6 by 6 10 cm grid overlaid on the SB. We model social interactions using a Hidden Markov Model which takes into consideration the relative trajectories and distances between pairs of mice and determines who initiates a contact, its progression and its properties using a simple topology states (idle, approach/avoid, contact, follow/avoid, described in detail in *Forkosh et al., 2019*).

Social dominance was assessed using the David's Score (DS), a measure based on the pairwise directionalities and numbers of agonistic interactions in a group (*David, 1987*). Chases during the four days of the baseline period were used to build the DS, which was then normalized to group number (*n* = 4), creating a continuous range between 0 (least dominant) and 3 (most dominant). The steepness of the social hierarchy was characterized as described in *de Vries et al., 2006* by using the slope of a line fitted to the DS from a ranked DS using Ordinary Least Squares regression. We used an implementation of this procedure made available in the open-source 'steepness' R package (*Leiva and de Vries, 2014*), whose output ranges between 0 and 1, with one meaning a very steep hierarchy in which power is unequally distributed between dominant and subordinate individuals. Despotism was defined as the fraction of the group's total number of chases that were initiated by the highest-ranking individual. Its values range as well between 0 and 1, in which one represents the presence of a very strong alpha who initiate all chases. Directional consistency was calculated using the average fraction of pairwise social interactions that occur in the direction from the individual who displayed more instances of an agonistic behavior to the individual who displayed fewer instances (*van Hooff and Wensing, 1987*; *Williamson et al., 2016*). A directional consistency equal to one indicates that all agonist interactions are directed from an individual with a higher DS to one with a lower DS. Finally, we used Landau's modified *h'* to assess the linearity of a social hierarchy, as described in *de Vries, 1995*, in which hierarchies which are fully linearly ordered are assigned a value of 1. We calculated both directional consistency and Landau's modified *h'* using functions made available in the R package 'compete' (*Curley et al., 2015*).

## Acute restraint

Before the beginning of the fifth night in the SB, mice were removed from the SB and restrained in a ventilated tube for 15 min. To account for the smaller size of females, we employed a smaller sized ventilated tube to ensure the same degree of movement restriction between sexes. At the end of the acute restraint, groups of mice were put back in their original SB and tracked for additional 12 hr.

## CMS protocol

Two separate batches of mice were exposed to three weeks of CMS prior to the behavioral test battery. A random combination of two stressors per day (one in the a.m. and one in the p.m. hours) was chosen among the followings: acute restraint in the dark (15 min), acute restraint in bright light (15 min, ~200 lux), acute restraint witnessing (half of the group at a time was restrained and placed inside the cage, 15 min each), removal of nesting material (24 hr), cage-tilt 30° along the vertical axis (6 hr), no bedding or nesting material (8 hr), wet bedding (6 hr), water avoidance (15 min), cage change (fresh cage every 30 min for a total of 4 hr), cage switching (mice are assigned the cage of another group of the same sex), overcrowding (eight mice per cage, 1 hr). For the water avoidance stress, an empty rat cage (395 × 346 cm) was filled with room temperature water. Mice were placed on a platform (10 × 12 cm), 2 cm above the water level, for 15 min.

On days 1, 3, 7, 10, 14, 17, and 21 both CMS and control mice were weighed. During the weighing session, their coat state was scored on a scale 0 to 3 according to the following criteria:

1. Bright and well-groomed coat. Clean eyes. No wounds.
2. Less shiny and less groomed coat OR unclean eyes. No wounds.
3. Dirty and dull coat and/or small wounds and not clear eyes.
4. Extensive piloerection OR alopecia with crusted eyes OR extensive wounds.

Cumulative coat state was calculated as the sum of the seven daily scores.

Control mice were kept in an adjacent room to the stressed mice and handled twice per week to obtain weight and coat scores.

## Behavioral battery

The day after the last stressor, mice started a behavioral test battery consisting of the OFT, 2-hr SPT, grouped SPT, the splash test (SPL), the nest building test (NBT), the EPM, a grouped sucrose preference, and the TST. Throughout the testing period, mice were maintained in their original groups and habituated to the testing room for at least one hour prior the start of the test. Forty-eight hours after the last test, mice were terminally anesthetized in isoflurane and sacrificed. Terminal bodyweight, plasma, adrenal glands, and thymus were collected. Adrenal glands and thymus were cleaned from fat tissue and weighed. Absolute values were adjusted to bodyweight using the bodyweights collected on day 1. Tissue weighing, corticosterone measurement, and behavioral scoring were performed by an experimenter blind to sex, condition, and social rank.

## Open field test

On the day following the last stressors (day 22), mice locomotor activity and exploratory behavior were assessed in the OFT for 10 min. The apparatus consisted in round arenas (diameter 38 cm) made of black polyvinylchloride (PVC) under dim illumination (3 lux). Mice were automatically tracked with ANYmaze Video Tracking System 6.13 (Stoelting, IL, USA). The space was virtually divided in an inner zone (diameter 16 cm) and an outer zone. Total distance traveled, distance from the center, speed, and turn angle were calculated across the full 10 min. In addition, distance traveled, speed, visits, and time spent in each of the subdivisions were used as parameters. Preference was calculated as follows: $\frac{outer\ zone\ time}{inner\ zone\ time}$.

## Two-hour daily sucrose preference test

Twenty-four hours after the OFT, the anhedonia phenotype was tested with a modified version of the SPT. Each group was assigned a test cage containing one water bottle and one bottle with 2% sucrose. One mouse per group at a time was placed in the test cage for two hours, across three consecutive days during the light phase (days 23, 24, and 25). At the end of each session, the bottles were weighed. At the end of the test the amounts of water and sucrose consumed were summed across the three sessions. Sucrose preference was calculated as $\frac{sucrose}{water+sucrose} * 100$.

## Grouped sucrose preference test

On day 27, sucrose preference was tested at a group level. Each group was given a bottle of water and a bottle of 2% sucrose within their home-cage. Their sucrose preference was calculated after 24 hr as above. A grouped sucrose preference value was obtained for each group.

## Splash test

On day 24, during the dark period, mice were tested in the splash test under dim light (3 lux). Mice were placed in their test cage for 5 min prior being sprayed on their dorsal coat twice (approximately 1 ml) with 10% sucrose solution. Mice were recorded for 5 min and total time spent grooming, and latency to the first grooming bout was manually scored using Solomon Coder 17.03.32 (https://solomon.andraspeter.com/).

## Nest building test

During the third day of the 2-hr sucrose preference, mice in the test cage were given a small square cotton pad of approximately 23 g. The cotton pad was weighed at the beginning of the test and at the end of the two hours and the percentage of intact material was calculated. The built nest was scored from 0 to 4 according to the following criteria:

1. Material untouched.
2. Material partially torn (50–90% remaining intact).
3. Material mostly shredded but often no identifiable nest site/scattered around.
4. Material accumulated in an identifiable nest site, but the nest is flat.
5. A (near) perfect nest: material fine shredded, doughnut like with walls higher than the mouse.

For nests matching only partially the description (e.g., identifiable flat nest, but less than 50% of torn material), half points were assigned.

### Elevated plus maze

On day 26, during the light phase, anxiety phenotype was assessed using the EPM test. An apparatus composed of four arms made of gray polyvinylchloride (PVC), two open without walls, two enclosed by 14 cm walls and a central platform (5 × 5 cm) was used. The apparatus was placed 33 cm from the ground under dim illumination (3 lux). Mice were placed on the central platform facing the open arms and let free to explore the apparatus for 10 min. Mice were automatically tracked using ANYmaze Video Tracking System 6.13 (Stoelting, IL, USA). Number of entries in each arms, time, and distance were calculated. In addition, closed arm preference was calculated as $\frac{time\ in\ closed\ arms}{time\ in\ closed\ +\ time\ in\ open\ arms}$.

### Tail suspension test

Stress coping behavior was assessed using the TST on day 28. Mice were hung by their tail 50 cm above the surface and their behavior recorded for 6 min. Immobility was automatically scored using ANYmaze Video Tracking System 6.13 (Stoelting, IL, USA) and number of immobility episodes and total time immobile were used as parameters.

### Corticosterone assessment

At sacrifice, trunk blood was collected in EDTA-coated tubes. Blood was centrifuged at 1,000 g for 15 min at 4°C. Plasma was retrieved and corticosterone levels were measured using [$^{125}$I] radioimmunoassay kit (MP Biomedicals), according to the manufacturer's instructions.

### Data analysis

#### Quality control and outlier removal

Low-quality tracks from the SBs were labeled by estimating the number of large (>100 pixels) interruptions in the mouse trajectories as well as the fraction of time mice were spotted outside the nest. Tracks where a mouse had more than 200 trajectory interruptions or was found outside the nest for less than 2% of the total monitoring time were excluded from further analyses. Additionally, all the tracks of an individual were excluded in cases when more than two of the four baseline day recordings of a mouse did not pass quality control thresholds. Based on these criteria, the complete SB data of four mice was excluded. A single value from the corticosterone outcomes was labeled as outlier and removed (453.4 ng/ml, >3.5 standard deviations away from the mean of the appropriate sex and condition grouping). The results of the EPM test for one mouse and the ones from the splash test for two mice were excluded due to a technical recording failure and the nest building test 'percent intact' value for one mouse was lost due to experimental failure.

#### Statistical testing

All statistical analyses were performed in R version 4.0.2 assisted by the '*Tidyverse*' ecosystem of packages (**R Core Development Team, 2013**; **Wickham et al., 2019**). The tests employed for each specific analysis are reported in the Results section. All inverse rank-transformed behavioral outcomes (Blom transform, RNOmni R package) (**McCaw, 2019**) were adjusted for batch effects using the standardized residuals of a linear model with each variable of interest as outcome and batch as a factorial predictor. Figure panels were assembled with the help of the '*cowplot*' R package (**Wilke, 2019**). Outcome data distributions were tested for deviations from normality (Shapiro-Wilk test) and heteroscedasticity (Levene's test). Whenever normality was violated and the data could not be transformed to fit a normal distribution, non-parametric tests were employed. Violations of homogeneity of variances are reported with each test. As this was an exploratory set of experiments without an a priori hypothesis regarding the association between dominance, stress response, and sex, no power calculations were performed, and sample sizes were chosen based published work with CMS interventions.

#### Principal component analysis

Principal component analysis (PCA) was employed to explore the sources of variance in our multidimensional behavioral dataset. Specifically, we performed PCA using singular value decomposition on scaled and centered data from the behavioral and physiological outcomes following CMS. Prior

to decomposition, missing data points (for a maximum of two values per individual) were replaced with the median of the respective outcome. The principal components (PCs) obtained were ranked by the total amount of variance explained. The top 3 PCs, namely PC1, PC2, and PC3, contained most of the variance in our dataset and were thus used to investigate the effects of known variables (sex, stress condition, social dominance status). To evaluate the influence of social dominance we assessed the association between PC1 and the David's Score within each experimental group.

## Acknowledgements

We thank Yair Shemesh, Oren Forkosh, Markus Nussbaumer, and Chadi Touma for their assistance in establishing the 'Social Box' paradigm. Thanks to Jessica Keverne for English writing support and advice. We are very grateful to the reviewers of this manuscript for their insightful feedback. AC is the incumbent of the Vera and John Schwartz Family Professorial Chair in Neurobiology at the Weizmann Institute and the head of the Max Planck Society–Weizmann Institute of Science Laboratory for Experimental Neuropsychiatry and Behavioral Neurogenetics. This work is supported by: an FP7 Grant from the European Research Council (260463, AC); a research grant from the Israel Science Foundation (1565/15, AC); the ERANET Program, supported by the Chief Scientist Office of the Israeli Ministry of Health (3–11389, AC); the project was funded by the Federal Ministry of Education and Research under the funding code 01KU1501A (AC); I-CORE Program of the Planning and Budgeting Committee and The Israel Science Foundation (grant no. 1916/12 to AC); Ruhman Family Laboratory for Research in the Neurobiology of Stress (AC); research support from Bruno and Simone Licht; the Perlman Family Foundation, founded by Louis L and Anita M Perlman (AC); the Adelis Foundation (AC); and Sonia T Marschak (AC). SK and EB are supported by the International Max Planck School for Translational Psychiatry (IMPRS-TP).

## Additional information

### Funding

| Funder | Grant reference number | Author |
| --- | --- | --- |
| H2020 European Research Council | 260463 | Alon Chen |
| Bundesministerium für Bildung und Forschung | 01KU1501A | Alon Chen |
| Israel Science Foundation | 1565/15 | Alon Chen |
| Israel Science Foundation | 1916/12 | Alon Chen |
| Israeli Ministry of Health | 3–11389 | Alon Chen |
| Ruhman Family Laboratory for Research on the Neurobiology of Stress | | Alon Chen |
| The Perlman Family Foundation | | Alon Chen |
| The Adelis Foundation | | Alon Chen |
| Max-Planck-Gesellschaft | Open-access funding | Alon Chen |
| International Max Planck School for Translational Psychiatry | | Alon Chen |
| Sonia T. Marschak | | Alon Chen |
| Bruno and Simone Licht | | Alon Chen |

The funders had no role in study design, data collection and interpretation, or the decision to submit the work for publication.

## Author contributions

Stoyo Karamihalev, Elena Brivio, Conceptualization, Data curation, Formal analysis, Investigation, Methodology, Writing - original draft, Writing - review and editing; Cornelia Flachskamm, Rainer Stoffel, Data curation, Investigation; Mathias V Schmidt, Methodology, Writing - original draft, Writing - review and editing; Alon Chen, Conceptualization, Supervision, Funding acquisition, Writing - original draft, Writing - review and editing

## Author ORCIDs

Stoyo Karamihalev https://orcid.org/0000-0002-1774-1548
Elena Brivio https://orcid.org/0000-0002-6213-0973
Mathias V Schmidt http://orcid.org/0000-0002-3788-2268
Alon Chen https://orcid.org/0000-0003-3625-8233

## Ethics

Animal experimentation: All experiments were approved by and conducted in accordance with the regulations of the local Animal Care and Use Committee (Government of Upper Bavaria, Munich, Germany), under licenses Az.: 55.2-1-54-2532-148-2012, Az.:55.2-1-54-2532-32-2016 and ROB-55.2-2532.Vet_02-18-50.

## Decision letter and Author response

Decision letter https://doi.org/10.7554/eLife.58723.sa1
Author response https://doi.org/10.7554/eLife.58723.sa2

# Additional files

## Supplementary files

- Transparent reporting form

## Data availability

All data used to support the findings of this work and the code used in performing the analyses and producing the figures for this manuscript is freely accessible in a GitHub repository: https://stoyokaramihalev.github.io/CMS_Dominance/ copy archived at (https://archive.softwareheritage.org/swh:1:dir:31e8482879f9eb61cf9341e95ccca36991847ba3). The MATLAB-based mouse tracking system is available at https://en.bio-protocol.org/prep207.

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
