## [Decision Letter]

**Acceptance summary:**

This paper asks how sex and social hierarchy influence behavioral adaptations to chronic stress. The paper makes innovations in measuring mouse behavior in a social context and concludes that social hierarchy has a sex-specific influence on multiple behavioral outcomes after chronic mild stress. Thus this work will be of interest to those interested in stress and related conditions, and the influence of sex on behavior. As it is standard practice to ignore social hierarchy in rodent studies, this paper is also of importance to the study of animal behavior more broadly.

**Decision letter after peer review:**

Thank you for submitting your article "Social dominance mediates behavioral adaptation to chronic stress in a sex-specific manner" for consideration by *eLife*. Your article has been reviewed by three peer reviewers, and the evaluation has been overseen by a Reviewing Editor and Kate Wassum as the Senior Editor. The following individual involved in review of your submission has agreed to reveal their identity: Jaideep Singh Bains (Reviewer #1).

The reviewers have discussed the reviews with one another and the Reviewing Editor has drafted this decision to help you prepare a revised submission.

Summary:

Karamihalev et al. use a novel approach to understand social hierarchy in mice and then ask whether this has impacts how individuals are affected by experimental chronic stress paradigms. The foundation question is: Does social hierarchy influence behavioral adaptations to chronic stress in a sex-specific manner? The consideration of sex and social hierarchy and interactions therein is extremely important. As it is standard practice to ignore social hierarchy in rodent studies, this paper fills an important gap. To this end, the authors use an ingenious system for measuring individual mouse behavior within social groupings. Based on their observations, they conclude that social hierarchy is a predictor of multiple behavioral outcomes after exposure to chronic mild stress, but the consequences are opposite in males and females.

Essential revisions:

Overall, all reviewers were very excited by the study and supportive of the approach being utilized, highlighting its novelty and impact. It has been uniformly agreed upon that no further experimentation is required, but there are several issues with interpretation and description that require attention.

1) Overall locomotion and exploration in the home-cage appears to be linked to hierarchy in males, but not in females. The authors suggest this means that, although male and female male and female social dominance hierarchies have a similar structure, they have different consequences for overall behavior. It is unclear how the authors can untangle whether it is the hierarchy that drives the behavior or whether it is the behavior that leads to a position in the hierarchy. For example, one possibility is that males who are dominant (*α*) explore more. The other argument is that it is the males that explore more who become dominant. So the question then becomes whether their dominance allows them to exhibit this behavior or whether, by exhibiting this behavior, they establish dominance. This argument is not one of semantics. It goes to the very heart of the paper which attempts to establish causal links between social hierarchy and subsequent behaviors.

2) Were the mice returned to the Social Box between CMS stress episodes and subsequent behavioral tests? The Materials and methods state, in reference to the Acute restraint that 'At the end of the stressors, groups of mice were put back in their original SB and tracked for an additional 12 hours' and in reference to the CMS that 'Throughout the testing period, mice were maintained in their original groups,' so I assume they were either returned to the SB or retained in groupings but not in the SB? Figure 2A is not informative in this regard. This aspect of the design requires clarification. It is also pertinent to the question of whether additional data were obtained, post stress, in the SB, and how these data looked – given this would be a potentially very interesting additional set of correlative data.

3) With respect to methodology, there is some additional clarification required for other points. First, there are many independent variables that are not clearly defined (e.g., steepness, despotism, directional consistency), which make data interpretation difficult. Second, the authors should also provide the behaviors in the correlation structure and how they are computed (subsection “Male and female dominance hierarchies”, third paragraph) minimally in the supplement and not make the reader reference another paper.

4) The PCA used in Figure 3 is not well explained. It could be explained in the Materials and methods because it is important for the reader to understand the differences between PCA1, PCA2, PCA3.

5) In Figure 4, the authors make rank-specific comparisons. It is not clear why the middle ranks are combined. The *β* and the *γ* appear to be as different from each other as the *α* from the *β* and the *γ* from the *δ*. If *β* and γ were interchangeable then I could understand this lumping of groups. But this is not the case.

6) A key part of the Results is in the section titled, "Sex-specific effects of dominance on CMS outcomes." But based on the data reported, it is not clear that social context is the determinant of the response to chronic stress. There is a nice association here, that suggests a predictive value to social hierarchy – most reliably in the dominant and subordinate, but it is very difficult to conclude that the responses to chronic stress are a consequence of this hierarchy. In order to make this conclusion, the authors would have to find a way to disrupt the hierarchy and then ask whether this affects responses to chronic stress. Additionally, the correlation of PC1 scores with the DS was powerful and suggests stability of the hierarchy. But it would have been better to re-test the hierarchy after the CMS manipulation. Are there any other metrics that suggest stability was maintained? In the absence of these studies where the hierarchy was manipulated and outcomes were measured, the authors should temper their statements indicating causal relationships between social status hierarchy and the effects of chronic stress, or vice versa.

---

## [Author Response]

Essential revisions:Overall, all reviewers were very excited by the study and supportive of the approach being utilized, highlighting its novelty and impact. It has been uniformly agreed upon that no further experimentation is required, but there are several issues with interpretation and description that require attention.1) Overall locomotion and exploration in the home-cage appears to be linked to hierarchy in males, but not in females. The authors suggest this means that, although male and female male and female social dominance hierarchies have a similar structure, they have different consequences for overall behavior. It is unclear how the authors can untangle whether it is the hierarchy that drives the behavior or whether it is the behavior that leads to a position in the hierarchy. For example, one possibility is that males who are dominant (α) explore more. The other argument is that it is the males that explore more who become dominant. So the question then becomes whether their dominance allows them to exhibit this behavior or whether, by exhibiting this behavior, they establish dominance. This argument is not one of semantics. It goes to the very heart of the paper which attempts to establish causal links between social hierarchy and subsequent behaviors.

We would like to thank the reviewers for highlighting this very important point. We believe position in the social dominance hierarchy is itself an expression of at least one stable underlying trait and have shown some evidence that it is robust against changes in social context in male mice (see Forkosh et al., 2019). Nonetheless, the reviewers are correct in stating that we cannot discern if the behavioral differences associated with social dominance are a cause or consequence (or neither) of occupying a specific social rank. This would be an excellent direction for future research although addressing this set of questions rigorously is a large and non-trivial undertaking. We believe it would take a combination of well-designed brain circuit-level manipulations and/or perhaps taking advantage of the more unstable social hierarchies.

However, despite our inability to pinpoint exactly if behavioral differences are a consequence of social rank or if rank is a consequence of the behavioral characteristics of the individual, we are able to show that for some behaviors this connection appears to be influenced by pre-existing social status and often only exists in one sex and not the other.

We have made some adjustments in the text of the manuscript to avoid giving readers the impression that we believe the behavioral differences are a consequence of social rank:

“These correlations indicate that overall locomotion and exploration of the home environment may be more strongly connected to social rank in male groups, while being seemingly independent of social status in females. […] Altogether, these findings suggest that male and female social dominance hierarchies, despite having a similar structure, have different relationships to overall behavior.”

And in the Discussion section:

“Additionally, our data suggest that an individual’s position in the hierarchy carries different implications for overall behavior in each sex. […] These associations likely reflect territorial or patrolling behavior in males, which may be less relevant to female social hierarchies.”

2) Were the mice returned to the Social Box between CMS stress episodes and subsequent behavioral tests? The Materials and methods state, in reference to the Acute restraint that 'At the end of the stressors, groups of mice were put back in their original SB and tracked for an additional 12 hours' and in reference to the CMS that 'Throughout the testing period, mice were maintained in their original groups,' so I assume they were either returned to the SB or retained in groupings but not in the SB? Figure 2A is not informative in this regard. This aspect of the design requires clarification. It is also pertinent to the question of whether additional data were obtained, post stress, in the SB, and how these data looked – given this would be a potentially very interesting additional set of correlative data.

We apologize for the lack of clarity in the Materials and methods section on this point. We have attempted to remove the ambiguity by making changes to Figure 2A and the Materials and methods:

“The fur of all mice was marked using four different colors under mild isoflurane anesthesia and mice were left to recover for few days before the start of the experiment. […] On day 6, animals were removed from the Social Box and placed in their original cage under standard housing conditions for the rest of the experimental procedure (see “Chronic mild stress protocol” and “Behavioral battery” sections).”

The animals were returned to the SB only following the acute restraint test on day 5, whereas during and after the CMS procedure animals were maintained in groups in standard housing conditions. We agree that a further testing in the SB after the CMS would generate interesting additional data. We hope to be able to address this in the future.

3) With respect to methodology, there is some additional clarification required for other points. First, there are many independent variables that are not clearly defined (e.g., steepness, despotism, directional consistency), which make data interpretation difficult. Second, the authors should also provide the behaviors in the correlation structure and how they are computed (subsection “Male and female dominance hierarchies”, third paragraph) minimally in the supplement and not make the reader reference another paper.

We thank the reviewers for drawing our attention to this and understand the need for further clarification. To address this, we have now expanded our explanation of the variables related to social hierarchy.

“We further calculated several properties of male and female hierarchies to explore potential differences in their characteristics. […] We found that male hierarchies were steeper, more linear, more despotic, and had higher directional consistency than those of females. Interestingly, mice housed in larger groups show analogous relationships between sexes (Williamson et al., 2019).”

“Social dominance was assessed using the David’s Score (DS), a measure based on the pairwise directionalities and numbers of agonistic interactions in a group (David, 1987). […]We calculated both directional consistency and Landau’s modified h’ using functions made available in the R package “compete” (Curley et al., 2015).”

We have also added clarifications of how SB behaviors are computed in the Materials and methods section, although, due to the large number of readouts and the extensive background behind how they are computed, we are not able to provide the same level of detail here as we have in the paper where we initially described them (Forkosh et al., 2019). We recognize that it is an inconvenience for readers to have to track down the specifics of how each behavior is computed in a separate paper, however the amount of detail needed for a satisfying description of each readout would considerably expand the size of this manuscript. We hope that a simplified explanation, such as the one below, will satisfy most readers and would refer those who still require more detail to our previous published work.

“From the location data we inferred agonistic interactions as well a variety of other behavioral readouts as described in (Forkosh et al., 2019). […] We model social interactions using a Hidden Markov Model, which takes into consideration the relative trajectories and distances between pairs of mice and determines who initiates a contact, its progression and its properties using a simple topology states (idle, approach/avoid, contact, follow/avoid, described in detail in Forkosh et al., 2019).”

4) The PCA used in Figure 3 is not well explained. It could be explained in the Materials and methods because it is important for the reader to understand the differences between PCA1, PCA2, PCA3.

We apologize for this oversight and have attempted to better explain PCA itself and the resulting projections. The following is now part of the designated PCA section in our Materials and methods section:

“Principal component analysis (PCA) was employed to explore the sources of variance in our multi-dimensional behavioral dataset. […] To evaluate the influence of social dominance we assessed the association between PC1 and the David’s score within each experimental group.”

5) In Figure 4, the authors make rank-specific comparisons. It is not clear why the middle ranks are combined. The β and the γ appear to be as different from each other as the α from the β and the γ from the δ. If β and γ were interchangeable then I could understand this lumping of groups. But this is not the case.

We are especially grateful to the reviewers for this comment. Upon reanalysis with the increased sample size, we did indeed find significant stability also in the *β* and *γ* ranks in males, making it especially clear that combining the ranks is unjustified. However, as stated above, we lack sufficient power to perform a rank-by-rank comparison within each sex to assess the social rank-specific effects of CMS. With that in mind, we have removed the comparisons that previously made up Figure 4 as well as the corresponding Supplementary Figure 4. Instead, we provide a new supplementary figure (Figure 3—figure supplement 2) that shows the correlations between social dominance as a continuous score (via the David’s score) and each individual behavioral or physiological readout within each sex and stress condition. We believe this presents a more complex, but also more accurate picture of the complex association patterns between social dominance, stress, and sex.

6) A key part of the Results is in the section titled, "Sex-specific effects of dominance on CMS outcomes." But based on the data reported, it is not clear that social context is the determinant of the response to chronic stress. There is a nice association here, that suggests a predictive value to social hierarchy – most reliably in the dominant and subordinate, but it is very difficult to conclude that the responses to chronic stress are a consequence of this hierarchy. In order to make this conclusion, the authors would have to find a way to disrupt the hierarchy and then ask whether this affects responses to chronic stress. Additionally, the correlation of PC1 scores with the DS was powerful and suggests stability of the hierarchy. But it would have been better to re-test the hierarchy after the CMS manipulation. Are there any other metrics that suggest stability was maintained? In the absence of these studies where the hierarchy was manipulated and outcomes were measured, the authors should temper their statements indicating causal relationships between social status hierarchy and the effects of chronic stress, or vice versa.

We thank the reviewers for this thoughtful comment and agree that establishing a clear causal link between social status and stress response would require further experimentation. We believe two explanations of the relationship between the social status and stress response are conceivable. Either, as discussed by the reviewers, hierarchical status directly modulates and determines an individual’s response to chronic stress, or there is a set of underlying characteristics that influences an individual’s social status and their behavioral response to chronic stress. In this scenario, prolonged occupation of a specific social rank does not directly alter stress response, but rather is itself an indication of the hidden feature(s) that govern both social dominance status and stress response.

Testing the social status of the mice a second time, following the CMS manipulation, would open a very interesting line of investigation and we did consider including it as part of our original design. However, by doing so we would have introduced a new and different set of questions related to the effects of CMS on social hierarchies (as well as on a variety of behaviors in the SB) and whether those effects depend on sex. These questions are of great interest and deserve thorough experimentation, but are outside the scope of the current work. Instead, here we decided to focus on exploring sex differences in social hierarchies in general and the relationship between sex, social dominance prior to stress exposure, and the behavioral response to chronic stress. The downside to this focus is that we cannot exclude the possibility that social hierarchies are systemically altered or otherwise disrupted by stress exposure. We have no evidence that the hierarchy status we observed at the start of the experiment was maintained. Nevertheless, our work demonstrates that a link exists between preexisting social status, sex, and chronic stress. Further, more detailed inquiries are needed to determine how exactly these three factors interact.

As with Point 1, we have made modifications to the text in order to clarify the limitations of the current work regarding the causal effects of social dominance.